# Total Dissolved Solids (TDS) Less Than 1000 ppm in Drinking Water Did Not Impact Nursery Pig Performance

**DOI:** 10.3390/vetsci9110622

**Published:** 2022-11-08

**Authors:** Ryan Samuel

**Affiliations:** Department of Animal Science, South Dakota State University, Brookings, SD 57007, USA; ryan.samuel@sdstate.edu; Tel.: +1-605-688-5165

**Keywords:** water quality, nursery pig, growth, swine, sulfates

## Abstract

**Simple Summary:**

Drinking water with high concentrations of total dissolved solids from sulfate salts can reduce animal performance (i.e., feed intake and efficiency of body weight gain) and water usage of newly weaned pigs through the nursery period. Total dissolved solids in drinking water may also negatively impact gut health (measured as intestinal permeability) of nursery pigs. Newly weaned pigs, approximately 20 days old, were stocked at 26 pigs per pen in 44 pens at the South Dakota State University wean-to-finish commercial research barn. Each pen was randomly assigned to receive one of four water treatments. When the concentration of total dissolved solids from sulfate salts is above tolerable levels, this can lead to piglets experiencing diarrhea. However, the sulfate salt levels that were supplied to newly weaned nursery pigs had no apparent impact on feed or water intake nor growth performance in this experiment.

**Abstract:**

High concentrations of total dissolved solids (TDS) in water have been reported to increase the incidence of diarrhea and reduce nursery pig growth performance. The objective of this study was to investigate the effects of drinking water with high concentrations of TDS from sulfate salts on nursery pigs. Weaned pigs sorted to equalize gender were placed in 44 pens with 26 pigs per pen. One of four water treatments was randomly assigned to each pen: (1) combination of CaSO_4_, MgSO_4_, and NaSO_4_; (2) CaSO_4_; (3) MgSO_4_; (4) NaSO_4_. Access to water and feed (nursery diets fed in four phases) was unrestricted throughout. The weights of pens were measured, feed remaining on weigh days was calculated, and pen water meters were read on d 0, 7, 21, 35, and 42. Water treatment did not affect (*p* > 0.07) average daily gain, average daily feed intake, or gut integrity of pigs. Water disappearance tended to be greater (*p* = 0.10) in pens receiving the CaSO_4_ water compared to the combination treatment from d 21 to 35. The TDS concentrations from sulfate salts used in this experiment did not impact the growth performance or feed or water disappearance of newly weaned nursery pigs.

## 1. Introduction

Water is essential for life as a key component of many metabolic processes and physiological functions including cell turgidity, temperature regulation, and the movement of nutrients and wastes around the body [1]. There is continual turnover of body water, but the amount of water in the body is relatively constant at any particular age. Water consumed directly or indirectly as a component of feed contributes to water intake. Water lost through respiration, defecation, and urination are the major losses of water for swine. Therefore, water turnover drives the requirement for water because water intake must balance water loss to keep body hydration constant. Factors that increase water turnover will increase the water requirement [2]. 

Starting at birth and through the entire production cycle, pigs should be allowed unrestricted access to water. Although the water requirement for suckling pigs is recognized to be relatively low compared to other stages of production due to the intake of water from milk, the importance of water intake should not be ignored [2]. In newly weaned pigs, McLeese et al. [3] observed that early water usage (approximately the first 5 days) was independent of the factors that normally affect water requirement. Afterwards, however, pig water usage appeared to be consistent with the known influences of water requirement, namely growth and feed intake [2].

Typically, swine produced in the United States change production sites at least once during their production cycle. At weaning, piglets are transported from farrowing barns to nursery barns or wean-to-finish barns [4]. There are inevitable differences in water quality due to different water sources at different production sites [5]. Patience [6] identified some of the potential sources of contamination of drinking water, including “poorly designed, located, or constructed wells”. Groundwater sources from specific aquifers can be naturally contaminated due to local geology. These differences in water quality between production sites may influence the production efficiency by altering nutrient usage and/or health [7,8]. Furthermore, facility design and management affect water quality [9,10]. Water quality variability within the South Dakota swine industry was recently assessed [11].

One measure of water quality is the total dissolved inorganic matter known as total dissolved solids (TDS). The most common elements that contribute to water with high concentrations of TDS are calcium, magnesium, and sodium [12]. General recommendations indicate that water containing <1000 ppm TDS presents no risk to pigs and that water above the maximum level of 3000 ppm TDS should be avoided, especially for nursery pigs [13,14,15,16,17]. However, currently available information about the impact of TDS on nursery performance is inexact [2]. For example, dietary nutrient utilization as affected by water with high concentrations of TDS are known to be variable, depending on the mineral that contributes most significantly to the high concentration of TDS [6]. 

Newly weaned pigs are often subject to changes in intestinal structure due to reductions in feed intake [18,19]. Water with significant concentrations of TDS (i.e., between 1000 and 2999 ppm) have been associated with incidences of diarrhea, which may also impact the intestinal structure and function. Specifically, excessive intake of magnesium and sodium sulfate salts through drinking water are associated with incidences of diarrhea [6]. Water with high TDS concentrations (i.e., between 3000 and 4999 ppm) has been observed to cause temporary water refusal [2]. Reduced water intake may further exacerbate intestinal damage and negatively impact health and growth performance. The health of the gastrointestinal tract, as the largest immune organ and the organ responsible for nutrient uptake, is critical to the efficient growth of the pig [3]. Gut health can be defined in different ways, one of which is the ability of the gut wall to act as a barrier against pathogens and toxins. The permeability of the gut wall may be altered during incidences of compromised gut health, such as diarrhea.

Previous research has identified that an increased concentration of total dissolved solids (TDS) in water can increase the incidence of diarrhea, but reduce the growth performance of nursery pigs [3,20]. Oppositely, it has been observed that water with high concentrations of TDS did not reduce growth performance [15,21,22]. Although the specific minerals of concern for weaned pigs have not been fully indicated [2], water with high magnesium and sodium sulfates concentrations can result in piglets with osmotic diarrhea [23]. The intestinal nutrient absorptive capacity and gut wall integrity can be negatively affected by diarrhea such that the gut becomes more permeable to toxins and pathogenic microorganisms [18]. Therefore, the primary objective of this study was to investigate the effects of drinking water with high concentrations of TDS on the growth performance, efficiency of body weight gain, and water usage of newly weaned pigs during the nursery period. A secondary objective was to investigate the effect of TDS in drinking water on the gut health (measured as gut permeability) of nursery pigs.

## 2. Materials and Methods

All experimental procedures were reviewed and approved by the Institutional Animal Care and Use Committee at South Dakota State University (19-013E). 

### 2.1. Animals, Housing, and Management

Newly weaned pigs (n = 1144), approximately 20 d old, were stocked at 26 pigs per pen (3.1 × 6.9 m; approximately 0.82 m^2^ per pig) in 44 pens at the South Dakota State University wean-to-finish commercial research barn (Flandreau, SD, USA). Pigs were “gate cut” and placed into pens of equal sex shortly after arrival at the facility. All pigs were managed according to standard procedures for vaccinations and treatments, as necessary. All pigs removed from the trial due to mortality and/or morbidity (e.g., severe diarrhea or unthrifty) were documented. All incidences of treatments, including the reason, the duration, and the outcome were documented. Pens were provided supplemental heat from gas brooders over mats that partially cover the fully slatted floors during the nursery period. The barn operated on mechanical ventilation with temperature setpoints at 26.1, 23.3, and 24.4 °C for d 1, 15, and 29, respectively. 

### 2.2. Feeding and Water

Pigs were provided free access to water and feed throughout the trial. Each pen was fitted with two cup waters which were cleaned daily [24] and one five space dry feeder of 178 cm total length (SD Industries, Alexandria, SD, USA). Feed was delivered to each pen by a single M-Series FEEDPro system (Feedlogic by ComDel Innovation, LLC, Wahpeton, ND, USA) quantifying and recording the amount of feed delivered to each pen. Each individual pen was fitted with a meter to measure water usage for each pen from one of four independent water lines.

### 2.3. Experimental Design

One of four water treatments were randomly assigned to each pen in a randomized complete block design: (1) combination (CaSO_4_, MgSO_4_, NaSO_4_); (2) high calcium (CaSO_4_); (3) high magnesium (MgSO_4_); (4) high sodium (NaSO_4_). 

### 2.4. Water Treatments

For the combination treatment, calcium, magnesium, and sodium were added to contribute similarly to the final concentration. The drinking water TDS concentration delivered to the pens was based on a stock solution diluted at 1:128 and delivered to the appropriate pens using a total of eight identical injectors to provide water to both rooms of the facility. The current TDS concentration (previously measured to be 354 ppm) in the rural water source (Big Sioux Community Water System, Egan, SD, USA) for the barn was considered adequate for creation of stock solutions. Sulfate salts (CaSO_4_, MgSO_4_, and/or NaSO_4_) were added to 20 L buckets of water and mixed to prepare the stock solutions. Each water line used a fixed ratio injector with water motor and pump proportional to water flow such that consistent amounts of the stock solution were injected. The stock solutions were mixed daily to reduce settling of the sulfates out of solution. The water samples for the laboratory analysis provided in Table 1 were collected from pens furthest away from the source.

### 2.5. Pen Weights and Feed Remaining 

Entire pen weights were measured weekly with a scale capable of holding all pigs from a pen at one time. On weigh days, the distance from the top of the feed to the top of the feeder was measured and the remaining feed in each feeder was calculated according to a prepared calibration curve relating the distance measurement and the density of the feed. 

### 2.6. Intestinal Permeability 

Intestinal permeability was determined using combined sugar probes [25]. At 7–10 d post-weaning, one barrow per pen was selected from two complete blocks plus one pig from a different block and housed in one of nine separate kennels with a modified floor to allow for total urine collection. Pigs were provided feed and water from their respective pens for the collection period of 6 hours before returning to their original pen. Pigs chosen represented the average weight pig per pen and maintained the repetition of 11 pens per treatment (44 pigs total). Each pig was orally administered a solution that contained 5% of both lactulose and mannitol at 15 mL/kg before they were placed into the dog kennels and the ratio of lactulose:mannitol (L:M) in urine was determined as a measure of gut permeability [26].

### 2.7. Statistical Analysis

Pen was the experimental unit for statistical analysis and provided 11 reps/treatment. Data were classified according to the applied water treatment with pen as a random variable. Block was included as a covariate in the model when significant. Differences between means was declared when *p* < 0.05 and a tendency will be declared when 0.05 < *p* < 0.10.

## 3. Results

### 3.1. Water Treatments

The unadulterated rural water source was previously analyzed to contain 354 ppm TDS [11]. The combination, MgSO_4_, and NaSO_4_ stock solutions appeared to increase the TDS and sulfate concentrations of the water for the nursery pigs (Table 1). Alternatively, the CaSO_4_ stock solution did not appear to increase the TDS concentration, which is likely due to the limited solubility of CaSO_4_ in water [27].

### 3.2. Animal Performance

Water treatment did not affect (*p* > 0.07) the average daily gain (ADG), average daily feed intake (ADFI), feed efficiency (gain:feed), nor average daily water intake (ADWI) of nursery pigs (Table 2).

### 3.3. Intestinal Permeability 

Pigs represented the average weight of pigs in the pen and pens were chosen to maintain the repetition of 11 pens per treatment. However, due to inconsistent results, most likely associated with inconsistent delivery of the oral solution, 11 replications were dropped from further analysis (Table 3). The average lactulose, mannitol, and the lactulose:mannitol ratios in urine from nursery pigs was not affected by treatment.

## 4. Discussion

Growth performance, efficiency of body weight gain, and water usage of newly weaned pigs during the nursery period was not affected by the drinking water concentrations of TDS in this study. Likewise, there were no measured effects of TDS in drinking water on the gut permeability of nursery pigs.

The NRC (1998) and Carson [2,28] summarized previous findings and provided the recommendation that water with TDS <1000 ppm would not be a concern for pigs. The current research failed to achieve TDS concentrations in the drinking water >1000 ppm. This was due to the standard injectors at the facility which are capable of diluting each stock solution at 1:128 or 0.78%. The facility also has 3% injectors available; however, the water flow demands of nursery pigs was too low for those injectors to work correctly. The additions of the sulfate adulterants were limited due to solubility of each compound. The CaSO_4_ was the least soluble, followed by NaSO_4_, and MgSO_4_ [27,29,30]. Although the stock solutions were stirred daily, there was a chance for the sulfates to precipitate out of solution over time. Future research should incorporate continuous stirring to help maintain consistent stock solutions.

In contrast to artificially adulterated water as provided in this investigation, Lozinski et al. (2022) [21] provided specific water sources, which had been identified as poor quality, that was collected at the source farms, transported to their research facility, and provided to nursery pigs from dedicated water bladders. However, theses water sources also did not affect nursery pig health or performance. Anderson and Stothers (1978) [13] compared a total salt intake of 11,000 ppm saline treatment versus 5000 ppm for the control accounting for sodium chloride in the feed and mixed salts in the water. At the same time, the salinity, or TDS content of the water high in either sulfates or chlorides, was 6000 ppm and the control treatment was 125 ppm. Although saline water generally increased water intake, ADG, ADFI, and F:G were not affected by the levels of TDS. Anderson et al. (1994) [7] determined that metabolizable energy from the diet decreased as the TDS concentration increased from 0 to 8000 ppm, but there were no impacts on ADG. 

## 5. Conclusions

Concentrations of TDS in drinking water above tolerable levels can result in osmotic diarrhea in nursery pigs. However, the concentrations of TDS from stock solutions of the sulfate salts CaSO_4_, NaSO_4_, and MgSO_4_ provided adulterated water to newly weaned nursery pigs that did not appear to influence growth performance nor feed or water intake. Identification of the minerals of specific concern for weaned pigs and the potential long-term impacts of water with elevated concentrations of TDS requires additional investigation.

## Figures and Tables

**Table 1 vetsci-09-00622-t001:** The Total Dissolved Solids (TDS) and sulfate concentrations of the adulterated waters provided to nursery pigs within 11 pens for each treatment from: (1) combination of CaSO_4_, MgSO_4_, and NaSO_4_; (2) CaSO_4_; (3) MgSO_4_; or (4) NaSO_4_.

	Combination ^1^	CaSO_4_	MgSO_4_	NaSO_4_
TDS, ppm	893	370	599	950
Sulfate, ppm	537	151	343	559

^1^ contributions from CaSO_4_, MgSO_4_, and NaSO_4._

**Table 2 vetsci-09-00622-t002:** The average daily gain (ADG), average daily feed intake (ADFI), feed efficiency (gain:feed) and average daily water intake (ADWI) of nursery pigs within 11 pens for each treatment provided water adulterated with: (1) combination of CaSO_4_, MgSO_4_, and NaSO_4_; (2) CaSO_4_; (3) MgSO_4_; or (4) NaSO_4_.

	Combination ^1^	CaSO_4_	MgSO_4_	NaSO_4_	SEM	*p*-Value
**ADG, g/d**						
0 to 7 d	393	376	386	384	6	0.25
7 to 21 d	431	435	440	416	4	0.24
21 to 35 d	697	703	699	708	4	0.75
35 to 42 d	900	894	920	875	6	0.07
0 to 42 d	591	591	597	585	3	0.43
**ADFI, g/d**						
0 to 7 d	365	363	346	346	7	0.52
7 to 21 d	678	670	674	670	4	0.88
21 to 35 d	1005	1009	1014	1002	5	0.86
35 to 42 d	1305	1297	1325	1317	7	0.55
0 to 42 d	838	835	840	834	4	0.93
**Gain:Feed**						
0 to 7 d	1.08	1.04	1.14	1.13	0.01	0.35
7 to 21 d	0.64	0.65	0.65	0.63	0.01	0.20
21 to 35 d	0.69	0.70	0.69	0.70	0.01	0.45
35 to 42 d	0.69	0.69	0.69	0.66	0.01	0.10
0 to 42 d	0.78	0.77	0.79	0.78	0.01	0.44
**ADWI, L/pig/d**						
0 to 7 d	1.6	2.2	1.7	1.7	0.1	0.14
7 to 21 d	2.1	2.5	2.2	2.4	0.1	0.47
21 to 35 d	2.1	3.2	2.4	2.5	0.2	0.10
35 to 42 d	3.7	3.9	3.4	3.2	0.2	0.30
0 to 42 d	2.4	2.9	2.4	2.5	0.1	0.08

^1^ contributions from CaSO_4_, MgSO_4_, and NaSO_4._

**Table 3 vetsci-09-00622-t003:** The average lactulose, mannitol, and the lactulose:mannitol ratio in urine from nursery pigs provided water adulterated with: (1) combination of CaSO_4_, MgSO_4_, and NaSO_4_; (2) CaSO_4_; (3) MgSO_4_; or (4) NaSO_4_.

	Combination ^1^	CaSO_4_	MgSO_4_	NaSO_4_	SEM	*p*-Value
Replications	8	9	9	7		
Lactulose, mM	16.7	10.0	9.5	9.2	1.3	0.12
Mannitol, mM	53.0	34.3	41.2	38.4	4.7	0.56
Lactulose:Mannitol	0.36	0.43	0.31	0.31	0.04	0.71

^1^ contributions from CaSO_4_, MgSO_4_, and NaSO_4._

## Data Availability

All data are contained within the article or supplementary material.

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
