# Peer review of "Total Dissolved Solids (TDS) Less Than 1000 ppm in Drinking Water Did Not Impact Nursery Pig Performance"

_vetsci, 2022, doi:10.3390/vetsci9110622_

Round 1
Reviewer 1 Report
Manuscript title: The concentrations of suflate salts used here had no impact on the parameters that author tested. Please try to incorporate the conclusion into the manuscript title.
Manuscript type: brief communication is more appropriate.
In veterinary textbook, salinity is used more often than TDS, can you incorporate this term into the manuscript somewhere.
Table 1: list the concentration in "ppm' in parallel? In discussion (lines 157-171), you used ppm.
In Tables: the numbers are listed without standard deviation or standard error. Is this a regular practice for this type of study?
line 159-161: "the general recommendations", cite a reference here.
Author Response
Thank you for your review of the manuscript. The requested information and changes are noted below.
Manuscript title: The concentrations of suflate salts used here had no impact on the parameters that author tested. Please try to incorporate the conclusion into the manuscript title.
The title was changed to more reflective of the study: Total Dissolved Solids (TDS) Less Than 1000 ppm in Drinking Water Did Not Impact Nursery Pig Performance
Manuscript type: brief communication is more appropriate.
The manuscript introduction, materials and methods, and discussion have been expanded and additional references added to address the length of the paper.
In veterinary textbook, salinity is used more often than TDS, can you incorporate this term into the manuscript somewhere.
Salinity versus TDS has been referred to within the discussion.
Table 1: list the concentration in "ppm' in parallel? In discussion (lines 157-171), you used ppm.
The units have been changed in the table.
In Tables: the numbers are listed without standard deviation or standard error. Is this a regular practice for this type of study?
Yes, this is standard to provide only the overall SEM.
line 159-161: "the general recommendations", cite a reference here.
A series of references were inserted.
Reviewer 2 Report
Please see attached

Author Response
Thank you for the careful review of the manuscript. The corrections and additional information are indicated below.
L53 What is gate cut? I assume it is drafting? Please either explain or use a more globally generic term
The term "gate cut" refers to the collective actions of opening the gates of the overstocked arrival pens, allowing the desired number of animals to walk out of the pens, and then closing the gates once the desired number of animals per experimental pen was achieved.
L54: “shortly after arrival” is this hours, days? If not, immediately where were they kept? Please clarify the phrase.
All animals were overstocked into 3-4 arrival pens per sex as soon as they arrived into the facility. Then, the desired number of animals in equal sex ratio were placed into experimental pens within 4 hours after arrival to the facility.
L55: you recorded all treatments. Were there any water medications given? Antibiotics in water is common in some practices.
There were no water medications provided to this group of animals.
L74: Water treatments. You made up stock solutions and mixed them into ? individual tanks or were the water lines (cups) lead directly from buckets?
Also was there constant mixing in the larger volume of water? As these are solids (correct me if wrong) and such “heavy” it is likely that if the water was to become still or less motion, they would likely start to “settle” and such concentration throughout the water source would be different. Was this taken into account? Were water samples taken from each pen to see if over time the added concentration of solids translated to the pen side level?
Water treatments were prepared and mixed in 20 L buckets. Each water line used a fixed ratio injector with water motor and pump proportional to water flow such that consistent amounts of the stock solution were injected. The stock solutions were mixed daily to reduce settling of the sulfates out of solution. The water samples for the analysis provided in Table 1 were collected from pens furthest away from the source.
L89: one pig per pen was selected for permeability testing, was this always one sex? or how was this accounted for?
Only barrows were selected to reduce the chance of urinating outside of the collection pan.
L96: you would not be able to account for sex in the pen average growth factors with this design which would have been interesting however could you account for anything like litter effect or previous grouping effect? Gut function and structure has been shown to be largely impacted by birth sow/litter and suckling environment.
The weaned pigs were mixed before going onto the truck and further mixed at arrival, so it would not be possible to trace back to the birth sow.
L101: you mentioned in the materials and methods that pig removals, vaccinations and treatments were recorded. Were there any results from that? Diarrhea occurrence?
There were no differences in pig removals, vaccinations, treatments, or diarrhea occurrence.
Table 2:
- Add in sample size in either table head or with each group.
the average daily gain (ADG), average daily feed intake (ADFI), feed efficiency (gain:feed) and average daily water intake (ADWI) of nursery pigs from 11 pens each provided water with added TDS from: 1) combination of CaSO4, MgSO4, and NaSO4; 2) CaSO4; 3) MgSO4; or 4) NaSO4
- Bold ADG and similar heading so they stand out.
Done
- ADFI 21 to 35 d and 35 to 41 d are these values kg/d? if so you need to change units. And check SEM still g?
The values have been updated and the SEM checked.
Discussion: Your discussion reads well but sounds more like an in-depth introduction. I see little mention of your own results in comparison to the previous studies.
- Add initial sentence to summarise the main findings
- Add own results in comparison
- A What were the limitations of the study? Existing water line design? Level of TDS mixed
- Your levels were all below this initially did that impact the results? Discuss further.
The text has been moved to the Introduction and the Discussion provided according to the recommendations.
- L158: you mention that general recommendation is that water containing <1000 ppm presents no risk to pigs which needs a reference.
A series of references has been included to address this oversight.
Reviewer 3 Report
The topic of the paper is interesting. It should be noted that the data were presented at the ASAS Annual 2020 and partially published at the following link https://www.nationalhogfarmer.com/animal-health/nursery-pig-performance-impacted-total-dissolved-solids-water
The paper has numerous limitations that do not allow publication under these conditions:
- the introduction is very concise; on the contrary, the discussion chapter must be moved to the introduction
-
- - Material and methods have numerous shortcomings
- Chap 2.3 water treatments. The characteristics of the rural water source are not specified; how was the TDS measured? field instruments or laboratory equipment?
- Cap 2.4 is water consumption calculated only on the basis of the consumption of the stock solution administered with the medicators (line78)? it's not very clear
- chap 2.5: citation 8 could be replaced with a description of the method in the text (it would be a short text)
-
- Results
3.2 animal performance: overall data for the period are missing (0-42d); in particular they could have significantly influenced the ADWI parameter
3.3 intestinal permeability: results of 11 replications were dropped. For some groups (combinations and NaSo4) the replications dropped represent 1/3 of the data.
- Discussion
A discussion is totally missing
Author Response
Thank you for the careful review of the submitted manuscript and for the helpful comments. Please see the responses below.
The paper has numerous limitations that do not allow publication under these conditions:
- the introduction is very concise; on the contrary, the discussion chapter must be moved to the introduction
The discussion text has been moved to the introduction and a discussion has replaced the text.
- Material and methods have numerous shortcomings
Additional details have been added.
- Chap 2.3 water treatments. The characteristics of the rural water source are not specified; how was the TDS measured? field instruments or laboratory equipment?
The source TDS has been provided from a previous analysis. All samples were analyzed at the Water and Environmental Engineering Research Center (WEERC) is located in the Jerome J. Lohr College of Engineering at SDSU
- Cap 2.4 is water consumption calculated only on the basis of the consumption of the stock solution administered with the medicators (line78)? it's not very clear
Each pen was fitted with a meter to measure water usage for each pen on an individual basis from one of four independent water lines.
- chap 2.5: citation 8 could be replaced with a description of the method in the text (it would be a short text)
Intestinal permeability was determined using combined sugar probes [25]. At 7 - 10 d post-weaning, one barrow per pen was selected from two complete blocks plus one pig from a different block and housed in one of nine separate kennels with a modified floor to allow for total urine collection. Pigs were provided feed and water from their respective pens for the collection period of 6 hours before returning to their original pen. Pigs chosen represented the average weight pig per pen and maintained the repetition of 11 pens per treatment (44 pigs total). Each pig was orally administered a solution that contained 5% of both lactulose and mannitol at 15 mL/kg before they were placed into the dog kennels and the ratio of lactulose:mannitol (L:M) in urine was determined as a measure of gut permeability [26].
- Results
3.2 animal performance: overall data for the period are missing (0-42d); in particular they could have significantly influenced the ADWI parameter
The additional information has been inserted into the table
3.3 intestinal permeability: results of 11 replications were dropped. For some groups (combinations and NaSo4) the replications dropped represent 1/3 of the data.
A total of 11 replications were dropped, leaving an approximately equal number of observations across the four treatments.
- Discussion
A discussion is totally missing
Discussion text has replaced the original text.
Round 2
Reviewer 3 Report
The authors have answered my doubts, but one aspect remains unclear. If the rural water source had a TDS = 492ppm, the addition of CaSO4 decreased the concentration of TDS. Can the authors explain this result?
Author Response
The authors have answered my doubts, but one aspect remains unclear. If the rural water source had a TDS = 492 ppm, the addition of CaSO4 decreased the concentration of TDS. Can the authors explain this result?
Thank you for the question which lead to the discovery that the incorrect value had been inserted into the text. The TDS from a water sample collected during the trial should read 354 ppm.
